# Associations between SNPs in Intestinal Cholesterol Absorption and Endogenous Cholesterol Synthesis Genes with Cholesterol Metabolism

**DOI:** 10.3390/biomedicines9101475

**Published:** 2021-10-14

**Authors:** Maite M. Schroor, Fatma B. A. Mokhtar, Jogchum Plat, Ronald P. Mensink

**Affiliations:** Department of Nutrition and Movement Sciences, NUTRIM School of Nutrition and Translational Research in Metabolism, Maastricht University, 6200 MD Maastricht, The Netherlands; j.plat@maastrichtuniversity.nl (J.P.); r.mensink@maastrichtuniversity.nl (R.P.M.)

**Keywords:** genetic variants, genetics, cholesterol metabolism, cholesterol absorption, cholesterol biosynthesis, sterols, campesterol, sitosterol, lathosterol

## Abstract

Single nucleotide polymorphisms (SNPs) have been associated with cholesterol metabolism and may partly explain large inter-individual variability in intestinal cholesterol absorption and endogenous cholesterol synthesis rates. This cross-sectional study therefore examined whether SNPs in genes encoding for proteins involved in intestinal cholesterol absorption (*ABCG5, ABCG8,* and *NPC1L1*) and endogenous cholesterol synthesis (*CYP51A1*, *DHCR7*, *DHCR24*, *HMGCR*, *HSD17B7*, *LBR,* and *MSMO1*) were associated with intestinal cholesterol absorption markers (total cholesterol (TC) standardized campesterol and sitosterol levels), an endogenous cholesterol synthesis marker (TC-standardized lathosterol levels), and serum low-density lipoprotein cholesterol (LDL-C) concentrations in a European cohort. *ABCG5* (rs4245786) and the tag SNP *ABCG8* (rs4245791) were significantly associated with serum campesterol and/or sitosterol levels. In contrast, *NPC1L1* (rs217429 and rs217416) were significantly associated with serum lathosterol levels. The tag SNP in *HMGCR* (rs12916) and a SNP in *LBR* (rs12141732) were significantly associated with serum LDL-C concentrations. SNPs in the cholesterol absorption genes were not associated with serum LDL-C concentrations. SNPs in *CYP51A1, DHCR24, HSD17B7,* and *MSMO1* were not associated with the serum non-cholesterol sterols and LDL-C concentrations. Given the variable efficiency of cholesterol-lowering interventions, the identification of SNPs associated with cholesterol metabolism could be a step forward towards personalized approaches.

## 1. Introduction

Cholesterol homeostasis is determined by the interaction between various complex processes including intestinal dietary and biliary cholesterol absorption, and endogenous cholesterol synthesis [1,2]. For the uptake of sterols into the enterocyte, the apical transporter Niemann-Pick C1-Like 1 (NPC1L1) plays a key role [3]. After absorption, the sterol efflux pump ATP-binding cassette (ABC) transporters G5 and G8 secrete a fraction of these sterols back into the intestinal lumen, while the remaining part is incorporated into chylomicrons and secreted into the circulation (Appendix A) [4]. De novo cholesterol synthesis, which involves approximately 30 reactions and more than 20 different enzymes, mainly takes place in the liver [2]. Other tissues, however, synthesize cholesterol as well [2]. The endogenous cholesterol synthesis pathway starts with acetyl-CoA, which is converted into the intermediate lanosterol in a multistep process. Lanosterol is ultimately converted into cholesterol via either the Bloch or the Kandutsch–Russell pathway (Appendix A). The intermediates in these two pathways differ, but the same enzymes are involved [5,6,7,8]. To estimate fractional intestinal cholesterol absorption, cholesterol-standardized campesterol and sitosterol levels can be used, while those of the Kandutsch–Russell pathway intermediate lathosterol reflect endogenous cholesterol synthesis rates. The use of these markers has been validated by correlating their plasma levels to stable isotope tracer measurements [9].

A reciprocal relation exists between intestinal cholesterol absorption and endogenous cholesterol synthesis [10]. For example, statin treatment decreases cholesterol synthesis but increases cholesterol absorption [11], while ezetimibe treatment results in the opposite effects [12]. Furthermore, large inter-individual differences are present in relative intestinal cholesterol absorption and endogenous cholesterol synthesis rates. To illustrate, intestinal cholesterol absorption values ranged from approximately 29% to 80% in healthy adults. However, within subject-variability was small [13]. For the cholesterol synthesis marker lathosterol, an intra-individual variation of around 23% and an inter-individual variation of more than 50% has been reported for healthy adults [14]. Genetic variants, including single-nucleotide polymorphisms (SNPs), might at least partly explain these large inter-individual variations and the wide ranges between individuals in responses to lipid-lowering medications [15]. In fact, some SNPs in intestinal cholesterol absorption genes have already been associated with fractional cholesterol absorption rates [16,17,18,19]. Additionally, several studies have reported associations between SNPs in genes related to intestinal cholesterol absorption and endogenous cholesterol synthesis with lipid-lowering effects of both pharmacological [20,21,22,23] and dietary interventions [24,25]. However, whether these associations relate to differences in intestinal cholesterol absorption and endogenous cholesterol synthesis rates has unfortunately not been documented. Identification of SNPs associated with intestinal cholesterol absorption and endogenous cholesterol synthesis is important, as findings may contribute to the development of personalized interventions aimed at improving cholesterol metabolism. The present study therefore investigated in a European population the relation between a number of selected SNPs in genes essential in intestinal cholesterol absorption—*ABCG5*, *ABCG8*, and *NPC1L1*—and SNPs in genes involved in endogenous cholesterol synthesis—*CYP51A1*, *DHCR7*, *DHCR24*, *HMGCR*, *HSD17B7*, *LBR*, and *MSMO1*—with serum intestinal cholesterol absorption markers (total cholesterol (TC) standardized levels of campesterol and sitosterol), an endogenous cholesterol synthesis marker (TC-standardized levels of lathosterol), and LDL-C concentrations.

## 2. Materials and Methods

### 2.1. Study Population

The present study included participants’ baseline data from five human intervention studies (Study 1 to Study 5), performed between 1997 and 2012 at Maastricht University, the Netherlands. All participants were recruited from Maastricht and the surrounding area, and data from N = 456 were available for the present study. Overall, the study sample consisted of healthy adults aged ≥18 years old. The body mass index (BMI) was calculated for each participant by diving their body weight (kg) by the square of height (m). Most participants had a normal weight (N = 225; 49.3%) or were overweight (N = 179; 39.3%). BMI of few participants fell within the underweight (N = 7; 1.5%), obesity class I (N = 28, 6.1%) or obesity class II (N = 6; 1.3%) range [26]. None of the participants used medication known to affect lipid metabolism. Details of the studies have been published [27,28,29,30], except for Study 4, which was a 6-week randomized, double-blinded, placebo-controlled parallel trial evaluating effects of plant-sterol ester supplementation as part of a combined lifestyle intervention. For the analysis of this project, we only used samples that were collected at baseline or at the end of a control period. All studies were approved by the Medical Ethics Committee of Maastricht University and were conducted according to the principles laid down in the Declaration of Helsinki. Written informed consent was obtained from all participants.

### 2.2. Blood Sampling and Biochemical Measurements

Blood samples were drawn from participants after an overnight fast. At least one hour after venipuncture, serum was obtained by centrifugation at 2000× *g* for 15–30 min at 4 °C and aliquots were stored at −80 °C. The concentrations of TC (CHOD/PAP method; Roche Diagnostics Systems Hoffmann-La Roche Ltd., Basel, Switzerland), high-density lipoprotein cholesterol (HDL-C) (precipitation method by adding phosphotungstic acid and magnesium ions, and CHOD/PAP method; Roche Diagnostics Systems Hoffmann-La Roche Ltd., Basel, Switzerland) and triacylglycerol (TAG) corrected for free glycerol (GPO-Trinder; Sigma Diagnostics, St Louis, USA) were determined in serum by using enzyme-based methods. LDL-C concentrations were calculated using the Friedewald equation [31].

Serum concentrations of the intestinal cholesterol absorption markers campesterol and sitosterol, and the endogenous cholesterol synthesis marker lathosterol were analyzed using gas chromatography with flame-ionization detection (GC-FID) in Study 1 and Study 5, while GC-mass spectrometry (GC-MS) was used in the three other studies. Further details on the non-cholesterol sterol analysis have been presented in the article by Mackay et al. [32]. Campesterol, sitosterol, and lathosterol concentrations are transported in plasma by cholesterol-rich lipoproteins, and therefore their concentrations were corrected for the differing number of lipoprotein particles by standardizing the concentrations of the markers to the TC concentrations (10^2^ × µmol/mmol TC) as measured with the CHOD/PAP method.

### 2.3. DNA Extraction, Genotyping, and Quality Control

Genomic DNA was isolated from either full blood or buffy coats using the QIAamp genomic DNA isolation kit (Westburg BV, Leusden, the Netherlands) according to the instructions of the manufacturer. After isolation, the purity of the genomic DNA was checked by measuring the 260/280 nm and the 260/230 nm ratios (NanoDrop; ND-1000 spectrophotometer, Isogen Lifescience B.V., De Meern, The Netherlands). For all samples, ratios varied between 1.7 and 1.9 and around 2.0, respectively. DNA concentrations were calculated using the relationship that an A_260_ of 1.0 corresponds with 50 µg/mL DNA. All samples were stored at −80 °C after isolation. After thawing, the quality of about 5% of the samples was tested by evaluating the degradation of DNA on agarose gels before further analysis. Results indicated that the quality of these samples was sufficient for genotyping. In the end, 471 DNA samples were genotyped by using the Axiom^TM^ Precision Medicine Research Array (PMRA) Kit (Thermo Fisher Scientific, Waltham, MA, USA) [33].

After running the arrays, the software package PLINK (version 1.90 beta; www.cog-genomics.org/plink/1.9/) [34] was used to exclude SNPs: (1) with >2% missing data, (2) located on sex chromosomes, (3) with a minor allele frequency (MAF) < 0.05, or (4) that deviated from Hardy–Weinberg Equilibrium (HWE) based on a *p*-value < 1 × 10^−10^. Six subjects were removed, because they had a heterozygosity rate ± 3 standard deviations (SDs) from the mean heterozygosity rate. Nine subjects were excluded because there was a sex discrepancy between DNA results with clinical records. Ultimately, 456 samples and 306,898 SNPs passed the quality-control criteria. Only SNPs in genes with a clear role in intestinal cholesterol absorption (*ABCG5, ABCG8,* and *NPC1L1*) or endogenous cholesterol synthesis (*CYP51A1, DHCR7, DHCR24, HMGCR, HSD17B7, LBR*, and *MSMO1*) that were present on the array and had passed the quality control steps were included in this study. An overview of the full gene names is provided in Appendix A. The rs-numbers of the selected SNPs are presented, except for two SNPs in *ABCG8* for which the rs-numbers were unknown. For these SNPs, their Affymetrix SNP ID (AX-number), i.e., their unique probe set identifier, is given. Appendix A presents information about these two SNPs that was provided by the PMRA array.

### 2.4. Statistics

Continuous values are reported as mean ± SD and categorical values as N (%). Visual inspection of histograms and Q-Q plots of the residuals showed a skewed distribution for TAG and concentrations were therefore log-transformed. Analysis of variance (ANOVA) was used to examine whether continuous variables differed significantly between the five studies. A chi-square test was used for categorical variables.

Possible deviations of the genotype frequencies from those expected under Hardy–Weinberg equilibrium (HWE) were assessed using chi-square tests in Microsoft Excel. Thereafter, SNPs with a genotype group with a frequency of <12 participants, which equals <2.5% of the sample size, were moved to the supplements. All SNPs in *DHCR7* were moved to the supplements due to this reason. Only for SNPs with a genotype group with a frequency of >12 participants, linkage disequilibrium (LD) was estimated and reported as r^2^-values for pairs of SNPs < 500 kB apart using the Haploview software package (version 4.1, Broad Institute of MIT and Harvard, Cambridge, MA, USA) [35]. A threshold of r^2^ ≥ 0.8 was used to define SNPs in LD. Haplotype blocks were constructed in Haploview by using the default algorithm as defined by Gabriel et al. [36]. In short, blocks were generated by this algorithm when at least 95% of the informative SNPs were in strong LD [36]. Furthermore, the Tagger program in Haploview version 4.1 was used to select tag SNPs using the pairwise tagging approach [35]. Selection criteria were a r^2^ threshold ≥ 0.8 and a log of the likelihood odds ratio (LOD) threshold of 3.0. Results of the statistical analysis of the tag SNPs are presented in the main text, whereas results for the captured SNPs have been placed in the Appendix A.

Linear regression analyses, corrected for the factor study, were used to examine associations among the TC-standardized non-cholesterol sterols and LDL-C concentrations. Additionally, the general linear model (GLM) was used to examine associations between the SNPs with serum non-cholesterol sterol levels, and LDL-C and TC concentrations. The analyses were adjusted for the factor study. In case of a statistically significant effect of a SNP, the differences in TC-standardized non-cholesterol sterol levels, serum LDL-C concentrations, or serum TC concentrations between the genotype groups were compared with a Bonferroni post-hoc test. The Benjamini–Hochberg multiple testing correction with a false discovery rate of 0.2 was applied to the GLM results for each gene separately. Only SNPs with genotype groups consisting of at least 12 individuals were included in the Benjamini–Hochberg correction. If the original *p*-value obtained from the general linear model analysis was smaller than the Benjamini–Hochberg critical value, the *p*-value was considered statistically significant. Next, for SNPs that were significantly associated with TC-standardized non-cholesterol sterols or LDL-C concentrations, an additive, dominant, or recessive multiple linear regression model was built with adjustment for the factor study. The additive model was used when the Bonferroni post-hoc test indicated that all three genotypes were significantly different or when the post-hoc test did not show which genotypes differed significantly. A dominant or recessive model was used when the Bonferroni post-hoc indicated a significant difference between only two genotypes. A dominant model was used if the least frequent homozygous genotype (e.g., aa) and the heterozygous genotype (e.g., aA) had a comparable relation with the outcome (i.e., the non-cholesterol sterols or LDL-C). The dominant model used the major homozygous group as reference, hence, AA was compared with aa + aA. Moreover, a recessive model was used if the least frequent homozygous genotype and the heterozygous genotype did not have a comparable relation with the outcome. The recessive model thus compared AA + aA with aa. All analyses were carried out using SPSS for Mac OS X (version 26.0, SPSS Inc., Chicago, IL, USA).

## 3. Results

Baseline characteristics for all participants and the five studies separately are shown in Appendix A. Significant differences between the studies were reported for all characteristics of the participants (all *p* < 0.05), except for gender (*p* = 0.064).

### 3.1. Associations between Markers for Cholesterol Absorption and Cholesterol Synthesis, and Serum LDL-C Concentrations

Linear regression analyses showed that, after controlling for the factor study, sitosterol was positively associated with campesterol (β = 1.39 × 10^2^ µmol/mmol TC; *p* < 0.001) and inversely with lathosterol (β = −0.09 × 10^2^ µmol/mmol TC; *p* = 0.025). In addition, campesterol showed a significant inverse association with lathosterol (β = −0.10 × 10^2^ µmol/mmol TC; *p* < 0.001). Campesterol, sitosterol, and lathosterol were not significantly associated with serum LDL-C concentrations (all *p* > 0.05) (Appendix A).

### 3.2. The Location and Allele Frequencies of the Selected SNPs

Appendix A shows the location and allele frequencies of the selected SNPs. The majority of SNPs were located in an intron and all SNPs had a call rate of ≥98.2%. The reference and alternative allele frequencies of the SNPs in our cohort were comparable to those of the European population, which were obtained from the National Center for Biotechnology Information (NCBI) [37]. Five of the 12 selected SNPs in the *ABCG8* gene (AX_11180448, rs41360247, rs4245791, rs4299376, rs6544713) deviated significantly from HWE (*p* < 0.05). All other SNPs were in HWE (all *p* > 0.05).

### 3.3. Linkage Disequilibrium and Tagging for SNPs in Genes Related to Intestinal Cholesterol Absorption

SNPs in *ABCG8* (rs4299376, rs6544713, and rs4245791) were in high LD (all r^2^ > 0.90) and consequently included in a haplotype block (Figure 1a). Haplotype block 2 included *ABCG8* (rs13390041, rs4077440, and rs3795860). Of these SNPs, rs13390041 and rs3795860 showed a high LD (r^2^ = 0.98). The tag SNP *ABCG8* (rs4245791) captured rs6544713 and rs4299376, while tag SNP *ABCG8* (rs3795860) captured rs13390041 (Table 1). For SNPs in *ABCG5* (Appendix A) and *NPC1L1* (Appendix A), no high LD was found (all r^2^ < 0.70).

### 3.4. Linkage Disequilibrium and Tagging for SNPs in Genes Related to Endogenous Cholesterol Synthesis

All SNPs in *HMGCR* were in (borderline) LD (all r^2^ ≥ 0.75) and consequently all SNPs were included in one single haplotype block (Figure 1b). One tag SNP in *HMGCR* was selected (rs12916), which captured rs12654264, rs3846662, and rs3846663 (Table 1). For *DHCR24*, rs6676774 and rs7551288 were in high LD (r^2^ = 0.90) and *DHCR24* (rs6676774) was selected as a tag SNP for rs7551288 (Appendix A; Table 1). None of the other SNPs in *DHCR24,* as well as the SNPs in *LBR* were in pairwise LD (all r^2^ < 0.80) (Appendix A).

### 3.5. Associations between SNPs in ABCG5, ABCG8, and NPC1L1 with TC-Standardized Serum Non-Cholesterol Sterol Levels and Serum LDL-C Concentrations

Significant associations were found for a SNP in *ABCG8* (rs4245791; *p* < 0.001) with both TC-standardized serum campesterol and TC-standardized serum sitosterol levels. *ABCG5* (rs4245786) was also significantly associated with TC-standardized sitosterol levels (*p* = 0.041). In addition, two SNPs in *NPC1L1* (rs217429 and rs217416) were significantly related with TC-standardized serum lathosterol levels (*p* < 0.05) (Table 2). After Benjamini–Hochberg multiple testing correction, all associations remained significant. Results for SNPs with a genotype group <12 participants are presented in Appendix A. A recessive model was built for *NPC1L1* (rs217429 and rs217416) with lathosterol levels (Appendix A). The additive models for *ABCG5* (rs4245786) with sitosterol, and for *ABCG8* (rs4245791) with sitosterol and campesterol levels can be found in Appendix A. No significant associations were observed between SNPs in *ABCG5*, *ABCG8,* or *NPC1L1* with serum LDL-C concentrations (all *p* > 0.05) (Table 2) or TC concentrations (all *p* > 0.05) (Appendix A).

### 3.6. Associations between SNPs in CYP51A1, DHCR24, HMGCR, HSD17B7, LBR, and MSMO1 with TC-Standardized Serum Non-Cholesterol Sterol Levels and Serum LDL-C Concentrations

None of the SNPs in genes essential in endogenous cholesterol synthesis showed a significant association with TC-standardized campesterol, sitosterol or lathosterol serum levels (all *p* > 0.05). Significant associations were reported for *HMGCR* (rs12916) and *LBR* (rs12141732) with serum LDL-C concentrations (all *p* < 0.05) (Table 3). Dominant models for these SNPs can be found in Appendix A. SNPs in *CYP51A1, DHCR24, HSD17B7,* and *MSMO1* were not significantly associated with serum LDL-C concentrations (all *p* > 0.05). Appendix A presents associations for SNPs with a genotype group <12 participants. Results for serum TC concentrations (Appendix A) are comparable to these of serum LDL-C concentrations (Table 3).

## 4. Discussion

Large inter-individual variation in intestinal cholesterol absorption and endogenous cholesterol synthesis exists, which may relate to differences in genetic background. Indeed, we found that SNPs in *ABCG5* and *ABCG8* were associated with intestinal cholesterol absorption, while SNPs in *NPC1L1* were significantly associated with endogenous cholesterol synthesis. However, none of the SNPs that were associated with intestinal cholesterol absorption or endogenous synthesis were associated with serum LDL-C concentrations, whereas SNPs in *HMGCR* and *LBR* did show such a relation. No associations were found for SNPs in *CYP51A1*, *DHCR24*, *HSD17B7*, and *MSMO1* with either one of the evaluated parameters.

*ABCG5* (rs4245786) was significantly related with TC-standardized serum sitosterol levels, a marker for intestinal cholesterol absorption. To the best of our knowledge, this association has not been reported before. *ABCG8* (rs4245791) had tagged rs6544713 and rs4299376, which all showed significant associations with intestinal cholesterol absorption markers. A previous study in a European cohort has also reported that SNPs in *ABCG8* were associated with cholesterol absorption [19]. In that study, the minor allele of rs41360247 was negatively related to cholesterol absorption and the minor allele of rs4245791 positively [19], which is in agreement with our findings.

For genes encoding enzymes of the endogenous cholesterol synthesis pathways, no significant associations with TC-standardized serum lathosterol levels were reported. Lathosterol is an intermediate in the Kandutsch–Russell pathway. To what extent the selected SNPs that are essential in endogenous cholesterol synthesis are associated with cholesterol synthesis rates in the Bloch pathway is not clear. For this, serum desmosterol should have been measured, which is specific for the Bloch pathway, whereas we analyzed lathosterol which is only part of the Kandutsch–Russell pathway. An explanation for the non-significant relations for the SNPs in the endogenous cholesterol synthesis genes that were selected in our study may be that other SNPs in these genes are associated with endogenous cholesterol synthesis, which were not included in the present study. Another explanation might be that the regulation of endogenous cholesterol synthesis is more complex and does not relate to one single SNP, as many enzymes are involved in the endogenous cholesterol synthesis pathway. In contrast to the absence of an association with lathosterol levels, SNPs in *LBR* (rs12141732) and *HMGCR* (rs12916) were significantly related with serum LDL-C concentrations. *HMGCR* (rs12916) was selected as tag SNP for *HMGCR* (rs12654264, rs3846662, and rs3846663), which also showed significant associations with serum LDL-C concentrations. For *HMGCR* (rs12654264, rs3846662, rs3846663, and rs12916) these associations with LDL-C concentrations agree with previous studies in Asian and European populations [38,39,40,41,42]. Although intestinal cholesterol absorption and endogenous cholesterol synthesis play a key role in the regulation of plasma LDL-C concentrations [2], they do not explain the significant associations between SNP in *HMGCR* and *LBR* with serum LDL-C concentrations. It is likely that other genes that are involved in cholesterol homeostasis have contributed to these findings.

Interestingly, SNPs in genes involved in intestinal cholesterol absorption were not exclusively associated with markers for their postulated physiological process. However, the cholesterol absorption genes *ABCG5*, *ABCG8,* and *NPC1L1* are not only expressed in the human intestine, but also in the liver [43,44]. On hepatocytes, *ABCG5/G8* regulates the secretion of cholesterol into bile and NPC1L1 facilitates hepatic cholesterol re-uptake, thereby finetuning an otherwise potentially large biliary and fecal loss of cholesterol [45]. In transgenic mice, overexpression of human *ABCG5* and *ABCG8* in the liver and small intestine reduced plasma plant sterol levels and fractional cholesterol absorption as measured by the fecal dual-isotope radio method [46]. In contrast, plasma lathosterol and liver mRNA levels of *HMGCR* were increased. Additionally, in vivo cholesterol synthesis was increased in the liver, possibly to compensate for the elevated biliary cholesterol secretion rates in these transgenic mice [46]. This animal study thus shows that *ABCG5* and *ABCG8* expression influences endogenous cholesterol synthesis which confirms our observations. Moreover, in our cohort, we noticed a similar association for an absorption gene, i.e., two SNPs in *NPC1L1* (rs217429 and rs217416) were associated with endogenous cholesterol synthesis. The question remains whether these associations between SNPs in intestinal cholesterol absorption genes and lathosterol only show the reciprocal phenomenon or should also be interpreted as a possible direct effect of the SNP on hepatic cholesterol synthesis. Temel et al. have shown that hepatic *NPC1L1* expression in transgenic mice increased hepatic cholesterol levels by enhancing the reuptake of cholesterol from the bile [47]. It may be that SNPs in *NPC1L1* have increased the expression or activity of NPC1L1 in the liver, which in turn impacts serum lathosterol levels. Furthermore, the SNPs in *ABCG5* and *ABCG8* that showed an association with intestinal cholesterol absorption were not associated with serum LDL-C concentrations and also did not show an inverse association with endogenous cholesterol synthesis. This may suggest that the cholesterol has been eliminated from the body, via for example hepatobiliary cholesterol excretion involving ABCG5/G8 or transintestinal cholesterol efflux [2,48].

There are some points that should be considered while interpreting our data. Firstly, it should be noted that almost all selected SNPs were located in intron regions. In general, SNPs in introns do not induce changes in protein-coding sequences, suggesting that they are potentially of less functional relevance than SNPs located in exons. However, SNPs in the intron regions can impact the protein via alternative regulation of splicing [49]. This can lead to incorrectly spliced mRNA, which may ultimately affect mRNA translation and result in non-functional proteins and can also have clinical consequences [50]. SNPs in introns could also serve as markers for other functionally relevant SNPs, as should be indicated by high LD between the SNPs. Secondly, significant differences were found between all baseline characteristics, except for gender distribution, between the five different studies. This heterogeneity between study populations was taken into account by correcting for the factor study in our analyses. In addition, only European individuals were included, which has further minimized this heterogeneity. In four studies, only individuals with a stable body weight (weight gain or loss of <3 kg for studies 1, 2 and 3 and <2 kg for study 5) could participate. For study 4, a stable body weight was not an inclusion criterion. It is therefore possible that some of the participants lost or gained some weight in the months preceding the study. However, it is not expected that possible changes in weight were related to a specific genotype group and therefore biased the results. Thirdly, this study had a relatively small sample size. This suggests that the significant findings that we found reflect strong associations. Our results can therefore help to determine whether individuals with specific genotypes are more sensitive to specific nutritional and pharmacological interventions, such as foods enriched with plant sterols or stanols, or ezetimibe and statin treatment. To illustrate, 4-week statin treatment in women with familiar hypercholesterolemia resulted in a significantly smaller percentage reduction in LDL-C concentrations in women with the AA genotype of *HMGCR* (rs3846662) compared to women with the other genotypes. Moreover, statin efficacy was significantly decreased in the AA group compared with women with the other genotypes [51]. This suggests that genotyping SNPs, even those located in the intron region, may play an important role in the development of more personalized treatment. Finally, an independent cohort in which we could replicate the positive findings was unavailable. Therefore, an additional study is needed to reach greater validity.

## 5. Conclusions

This study showed that several SNPs in genes that are essential in intestinal cholesterol absorption were associated with serum markers for intestinal cholesterol absorption and/or endogenous cholesterol synthesis. In addition, a number of SNPs in genes that are essential in endogenous cholesterol synthesis were associated with serum LDL-C concentrations in a European cohort.

## Figures and Tables

**Figure 1 biomedicines-09-01475-f001:**
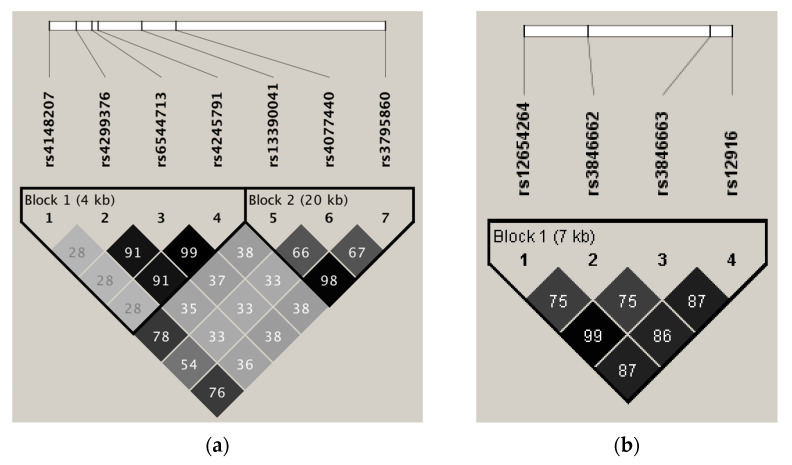
Pairwise LD among (**a**) 7 SNPs in *ABCG8* and (**b**) 4 SNPs in *HMGCR* is indicated in the diamond shapes. The triangles mark the two haplotype blocks within this region (based on the confidence interval of D’). The shading with a dark grey to white gradient indicates the level of higher to lower LD between each pair of SNPs based on the r^2^-value. The LD plot was created by Haploview version 4.1 [35].

**Table 1 biomedicines-09-01475-t001:** Tag SNPs and their captured SNPs with their corresponding r^2^-values.

Gene	Tag SNP	Captured SNP	R^2^-Value
*ABCG8*	rs4245791	rs6544713	0.995
	rs4245791	rs4299376	0.919
	rs3795860	rs13390041	0.982
*DHCR24*	rs6676774	rs7551288	0.906
*HMGCR*	rs12916	rs12654264	0.872
	rs12916	rs3846662	0.862
	rs12916	rs3846663	0.879

Tag SNPs and their captured SNPs were selected using the Tagger program within Haploview version 4.1. [35].

**Table 2 biomedicines-09-01475-t002:** Associations between various SNPs in cholesterol absorption genes with serum TC-standardized campesterol, sitosterol and lathosterol levels (N = 455), and serum LDL-C concentrations (N = 456).

Gene	SNP	Genotype	N	Campesterol 10^2^ × µmol/mmol TC		Sitosterol 10^2^ × µmol/mmol TC		Lathosterol 10^2^ × µmol/mmol TC		N	LDL-C mmol/L	
				Mean (95% CI)	*p*-Value	Mean (95% CI)	*p*-Value	Mean (95% CI)	*p*-Value		Mean (95% CI)	*p*-Value
*ABCG5*	rs4245786	AA	266	252 (236–267)	0.074	152 (142–162)	0.041 ^$^	120 (112–129)	0.959	266	3.44 (3.30–3.57)	0.306
		AG	160	230 (212–249)		136 (124–148)		120 (110–131)		161	3.34 (3.18–3.50)	
		GG	29	259 (222–296)		154 (130–178)		123 (103–144)		29	3.23 (2.90–3.55)	
	rs7599296	AA	15	261 (210–312)	0.228	164 (131–197)	0.173	109 (81–137)	0.653	15	3.40 (2.95–3.85)	0.980
		AG	141	255 (236–274)		152 (140–165)		119 (108–130)		141	3.38 (3.21–3.54)	
		GG	299	239 (224–254)		143 (133–152)		122 (113–130)		300	3.39 (3.26–3.52)	
	rs4148184	TT	74	232 (207–256)	0.297	142 (126–158)	0.803	117 (103–130)	0.217	74	3.30 (3.08–3.51)	0.561
		TC	219	251 (235–268)		148 (137–159)		117 (108–126)		219	3.42 (3.28–2.57)	
		CC	161	242 (223–260)		146 (134–158)		126 (116–137)		162	3.39 (3.23–3.55)	
	rs13396273	TT	53	236 (207–264)	0.431	144 (126–163)	0.819	116 (101–132)	0.526	53	3.36 (3.11–3.60)	0.922
		TC	214	251 (234–267)		148 (138–159)		119 (109–128)		214	3.40 (3.26–3.55)	
		CC	188	240 (222–257)		145 (133–156)		124 (114–134)		189	3.38 (3.22–3.53)	
*ABCG8*	rs4148207	TT	156	249 (231–268)	0.757	151 (139–163)	0.364	121 (111–131)	0.713	157	3.34 (3.18–3.50)	0.530
		TC	227	243 (226–259)		145 (123–155)		121 (112–130)		227	3.43 (3.29–3.58)	
		CC	72	241 (216–266)		139 (123–155)		116 (102–129)		72	3.35 (3.13–3.57)	
	rs3795860 ^+^	TT	128	253 (234–273)		154 (141–167)		120 (109–131)		129	3.32 (3.15–3.50)	
		TC	233	244 (228–260)	0.342	146 (135–156)	0.174	123 (114–131)	0.515	233	3.46 (3.32–3.60)	0.175
		CC	94	234 (211–257)		138 (123–152)		115 (102–127)		94	3.29 (3.09–3.49)	
	rs4077440	TT	92	256 (233–279)	0.129	154 (140–169)	0.125	120 (107–132)	0.378	92	3.38 (3.18–3.58)	0.252
		TC	217	249 (232–266)		149 (138–159)		124 (115–133)		218	3.45 (3.31–1.60)	
		CC	145	232 (213–251)		138 (126–150)		116 (105–126)		145	3.30 (3.13–3.46)	
	AX_82902928	--	197	248 (231–265)	0.752	151 (140–161)	0.334	120 (111–130)	0.955	197	3.40 (3.25–3.55)	0.145
		-AC	192	240 (223–258)		141 (130–165)		120 (110–130)		193	3.43 (3.28–3.58)	
		ACAC	66	246 (219–272)		147 (130–165)		122 (108–137)		66	3.19 (2.60–3.42)	
	rs4245791 ^+^	TT	206	221 (205–237) ^A^		130 (120–141) ^A^		123 (114–132)		206	3.32 (3.17–3.47)	
		TC	215	256 (239–272) ^B^	<0.001 ^$^	153 (143–164) ^B^	<0.001 ^$^	119 (109–128)	0.642	216	3.46 (3.31–3.61)	0.239
		CC	34	315 (282–349) ^C^		180 (176–219) ^C^		117 (97–136)		34	3.34 (3.04–3.65)	
*NPC1L1*	rs217429	AA	259	239 (223–254)	0.190	142 (132–152)	0.134	119 (110–128) ^A^	0.017 ^#^	259	3.37 (3.23–3.50)	0.825
		AC	169	256 (238–275)		154 (142–166)		117 (107–127) ^A^		170	3.42 (3.26–3.58)	
		CC	27	238 (200–276)		146 (121–170)		149 (128–170) ^B^		27	3.39 (3.06–3.73)	
	rs217416	TT	239	240 (223–256)	0.208	143 (132–153)	0.236	119 (110–127) ^A^	0.020 ^#^	239	3.40 (3.26–3.54)	0.922
		TC	189	254 (237–272)		153 (141–164)		118 (108–128) ^A^		190	3.38 (3.23–3.54)	
		CC	25	228 (188–267)		140 (114–165)		149 (128–171) ^B^		25	3.33 (2.98–3.67)	
	rs11763759	TT	208	244 (227–261)	0.961	145 (134–156)	0.938	120 (111–130)	0.953	209	3.42 (3.27–3.56)	0.084
		TC	202	246 (229–263)		147 (136–158)		120 (111–129)		202	3.31 (3.16–3.46)	
		CC	43	242 (211–273)		149 (128–169)		123 (106–140)		43	3.62 (3.35–3.89)	
	rs2072183	CC	18	260 (213–307)	0.314	154 (123–184)	0.361	121 (95–147)	0.862	18	3.33 (2.91–3.75)	0.930
		CG	173	254 (235–272)		152 (140–164)		122 (112–133)		174	3.40 (3.24–3.57)	
		GG	263	240 (225–255)		143 (134–153)		119 (111–128)		263	3.38 (3.25–3.52)	

Abbreviations: LDL-C = low-density lipoprotein cholesterol; SNP = single-nucleotide polymorphism; TC = total cholesterol. Note: All analyses were adjusted for the factor study. Data are presented as estimated marginal means (95% CI). Non-cholesterol sterol levels were missing for N = 1. Different letters between genotypes within a SNP indicate significantly different non-cholesterol sterol levels between the genotypes based on a Bonferroni post-hoc test. Significant *p*-values remained significant after adjustment for multiple testing by calculating critical values for each *p*-value using the Benjamini–Hochberg principle. ^+^ Indicates a tag SNP. ^#^ Recessive models are presented in the Appendix A. ^$^ Additive models are presented in the Appendix A.

**Table 3 biomedicines-09-01475-t003:** Associations between various SNPs in endogenous cholesterol synthesis genes with serum TC-standardized campesterol, sitosterol and lathosterol levels (N = 455), and serum LDL-C concentrations (N = 456).

Gene	SNP	Genotype	N	Campesterol 10^2^ × µmol/mmol TC		Sitosterol 10^2^ × µmol/mmol TC		Lathosterol 10^2^ × µmol/mmol TC		N	LDL-C mmol/L	
				Mean (95% CI)	*p*-Value	Mean (95% CI)	*p*-Value	Mean (95% CI)	*p*-Value		Mean (95% CI)	*p*-Value
*CYP51A1*	rs35968894	AA	161	240 (222–258)	0.239	142 (131–154)	0.334	115 (104–124)	0.066	161	3.40 (3.24–3.56)	0.976
		AG	223	241 (224–258)		146 (135–157)		127 (118–136)		224	3.38 (3.23–3.53)	
		GG	71	262 (238–287)		156 (140–172)		117 (103–131)		71	3.39 (3.17–3.60)	
*DHCR24*	rs6676774 ^+^	AA	75	231 (207–256)	0.436	144 (128–160)	0.887	120 (106–134)	0.535	75	3.42 (3.20–3.63)	
		AG	208	246 (230–263)		146 (135–157)		123 (114–132)		208	3.30 (3.16–3.45)	0.122
		GG	172	249 (230–267)		148 (136–160)		117 (107–127)		173	3.48 (3.33–3.64)	
	rs718265	AA	43	231 (200–263)	0.292	143 (123–164)	0.794	117 (98–134)	0.570	43	3.35 (3.07–3.62)	0.460
		AG	190	252 (235–269)		149 (138–160)		123 (114–133)		190	3.34 (3.19–3.49)	
		GG	222	240 (223–257)		145 (134–156)		118 (109–127)		223	3.44 (3.29–3.59)	
*HMGCR*	rs12916 ^+^	TT	151	240 (221–260)	0.373	145 (133–158)	0.541	122 (112–133)	0.838	152	3.22 (3.05–3.39) ^A^	0.011 ^@^
		TC	231	242 (226–259)		145 (134–155)		119 (110–128)		231	3.49 (3.35–3.63) ^B^	
		CC	73	259 (234–284)		154 (138–170)		122 (108–135)		73	3.35 (3.13–3.56)	
*HSD17B7*	rs77482353	AA	156	241 (222–259)	0.676	142 (130–154)	0.516	121 (111–131)	0.889	156	3.40 (3.24–3.56)	0.070
		AG	227	250 (233–266)		150 (139–160)		120 (111–130)		228	3.32 (3.18–3.47)	
		GG	68	246 (220–272)		150 (133–167)		117 (103–132)		68	3.60 (3.72–3.83)	
*LBR*	rs6678087	TT	141	247 (228–267)	0.367	147 (134–160)	0.988	120 (109–131)	0.997	141	3.41 (3.24–3.58)	0.970
		TC	223	248 (232–265)		147 (136–157)		121 (112–130)		223	3.39 (2.25–3.53)	
		CC	90	232 (209–254)		146 (131–161)		120 (108–133)		91	3.39 (3.19–3.59)	
	rs12141732	TT	226	241 (224–258)	0.706	144 (133–155)	0.453	121 (111–130)	0.799	227	3.50 (3.35–3.65) ^A^	0.027 ^@^
		TC	194	248 (232–265)		147 (136–158)		121 (112–130)		194	3.28 (3.13–3.43) ^B^	
		CC	34	251 (216–286)		159 (136–182)		114 (95–134)		34	3.50 (3.20–3.81)	
*MSMO1*	rs17046216	AA	53	237 (209–266)	0.112	147 (128–165)	0.347	113 (97–128)	0.542	53	3.63 (3.38–3.88)	0.101
		AG	205	236 (219–253)		142 (131–153)		121 (112–131)		206	3.35 (3.21–3.50)	
		GG	197	256 (239–273)		151 (140–162)		122 (112–131)		197	3.36 (3.21–3.51)	

Abbreviations: LDL-C = low-density lipoprotein cholesterol; SNP = single-nucleotide polymorphism; TC = total cholesterol. Note: All analyses were adjusted for the factor study. Data are presented as estimated marginal means (95% CI). Non-cholesterol sterol levels were missing for N = 1. Different letters between genotypes within a SNP indicate significantly different LDL-C concentrations between the genotypes based on a Bonferroni post-hoc test. Significant *p*-values remained significant after adjustment for multiple testing by calculating critical values for each *p*-value using the Benjamini–Hochberg principle. ^+^ Indicates a tag SNP. ^@^ Dominant models are presented in the Appendix A.

## Data Availability

All data are included in this manuscript.

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
