# Peer review of "Associations between SNPs in Intestinal Cholesterol Absorption and Endogenous Cholesterol Synthesis Genes with Cholesterol Metabolism"

_biomedicines, 2021, doi:10.3390/biomedicines9101475_

Round 1
Reviewer 1 Report
Although the study has several limitations, the results are interesting and well explained. It would be nice if the recent weight loss of the five analyzed populations could be reported as it appears to be associated with cholesterol absorption along with other metabolic factors including obesity, type 2 diabetes and fatty liver.
the presentation of the data could be more explanatory using the median value and the confidence interval. In this way the reader will be able to better appreciate how many overweight / obese patients are included (the figure seems to vary from 22.8 +/- 2.5 to 26.3 +/- 3.6). Body composition can also be interesting.
Reviewer 2 Report
This is a very interesting paper. The study is well designed and the conclusions fit with the data. This paper deserves publication on "Biomedicine"
Reviewer 3 Report
The proposed manuscript describes the relationship of cholesterol metabolism genes SNPs and cholesterol metabolism markers. The number of significant SNPs is surprisingly low, but reasons are well discussed. Still, novel findings are presented in the manuscript. The formal presentation of results is excellent.
I would recommend adding more details about non-cholesterol sterols analytical methods - what were the extraction procedures, were they same in all merged studies, possibly reference to analytical method.
